# Enantioselectivity of discretized helical supramolecule consisting of achiral cobalt phthalocyanines via chiral-induced spin selectivity effect

Hiroki Aizawa [1,2], Takuro Sato [1,2] ✉, Saori Maki-Yonekura [3], Koji Yonekura [3,4,5], Kiyofumi Takaba [3], Tasuku Hamaguchi [3,6], Taketoshi Minato [1] & Hiroshi M. Yamamoto [1,2] ✉

Enantioselectivity of helical aggregation is conventionally directed either by its homochiral ingredients or by introduction of chiral catalysis. The fundamental question, then, is whether helical aggregation that consists only of achiral components can obtain enantioselectivity in the absence of chiral catalysis. Here, by exploiting enantiospecific interaction due to chiral-induced spin selectivity (CISS) that has been known to work to enantio-separate a racemic mixture of chiral molecules, we demonstrate the enantioselectivity in the assembly of mesoscale helical supramolecules consisting of achiral cobalt phthalocyanines. The helical nature in our supramolecules is revealed to be mesoscopically incorporated by dislocation-induced discretized twists, unlike the case of chiral molecules whose chirality are determined microscopically by chemical bond. The relevance of CISS effect in the discretized helical supramolecules is further confirmed by the appearance of spin-polarized current through the system. These observations mean that the application of CISS-based enantioselectivity is no longer limited to systems with microscopic chirality but is expanded to the one with mesoscopic chirality.

A chiral object—a motif that cannot be superimposed on its mirror image (enantiomer)—is characterized by an enantiospecific response to a chiral field. In other words, the chiral field can recognize the handedness of the chiral object through the enantiospecific interaction with the object, enabling us in theory to preferentially select one of the two possible enantiomeric forms[1–10]. In this sense, a chiral field manifests itself as an enantioselective field. An ultimate goal in this context is a synthesis of a homochiral aggregation from achiral components purely through the chiral field, which is often referred to as

absolute enantioselection[11,12], but its experimental realization has been quite challenging.

Recently, a new enantioselective interaction in a quantum mechanical manner has been identified between chiral molecules and ferromagnetic spins. A heart of the enantioselectivity lies on the so-called chiral-induced spin selectivity (CISS) effect[13–15]; in chiral systems, an electron's charge motion engages with its spin angular momentum, the direction of which is symmetrically fixed to be parallel or anti-parallel to that of momentum depending on the

[1]Institute for Molecular Science, Myodaiji, Okazaki 444-8585, Japan. [2]the Graduate University for Advanced Studies, Myodaiji, Okazaki 444-8585, Japan. [3]Biostructural Mechanism Laboratory, RIKEN SPring-8 Center, Hyogo 679-5148, Japan. [4]Institute of Multidisciplinary Research for Advanced Materials, Tohoku University, 2-1-1 Katahira, Aoba-ku, Sendai 980-8577, Japan. [5]Advanced Electron Microscope Development Unit, RIKEN-JEOL Collaboration Center, RIKEN Baton Zone Program, 1-1-1 Kouto, Sayo, Hyogo 679-5148, Japan. [6]Present address: Institute of Multidisciplinary Research for Advanced Materials, Tohoku University, 2-1-1 Katahira, Aoba-ku, Sendai 980-8577, Japan. ✉e-mail: takurosato@ims.ac.jp; yhiroshi@ims.ac.jp

handedness (namely, spin-momentum locking). This feature immediately offers a situation where on approaching a substrate, the chiral molecule is subjected to time-dependent electrostatic force, which rearranges a charge distribution in the molecule, producing enantiospecific spin flow. In a closed system like molecule, the spin flow will be accumulated at the edge of the molecule. It should be noted that the spin accumulation by CISS accompanies not a single spin but a pair of anti-parallel spins at the two ends of a chiral system[16]. The formation of the latter is indispensable for the enantioselection because both chirality and the pair of anti-parallel spins can be classified into the same group of symmetry in that both of their signs are switched by space-inversion operation but not by rotational one, contrary to the case of a single spin or ferromagnetic spins whose signs become opposite by 180° rotation. Then, if the substrate shows out-of-plane ferromagnetic ordering, the emergent spin via CISS effect can be interacted with the ferromagnetic spin through short-ranged quantum exchange coupling, giving the enantioselectivity externally controllable by the direction of the ferromagnetic spins in the substrates (Fig. 1a). The field associated with the chiral object here can be expressed as the product of ferromagnetic spin, $S$, and time-derivative of electric field, $E$, namely $S \cdot dE/dt$, which changes its sign not under time-reversal but under space-inversion operation[17] ($T$-even pseudo scalar, or electrical toroidal monopole, $G_0$). This enantioselectivity based on CISS effect has been already shown to be able to separate a racemic mixture of helical amino acids, whose chirality is ab initio incorporated by their microscopic structures that can be coupled to the magnetic substrate via short-range exchange interaction, into its two enantiomeric components[13,18–20]. (Note that enantioselective electrochemical reactions are discussed in ref. 20). However, a critical open question is whether this CISS-based enantioselectivity is applicable to chiral superstructure consisting only of achiral components where only weak intermolecular interaction is expected to work as the origin of its chirality. The issue is intimately tied to a possible absolute enantioselection via the CISS effect.

An experimental platform accessible to this question is provided by a family of phthalocyanines (Pcs)[21]. Pcs are π-conjugated organic molecules whose molecular structure is characterized by an inversion-

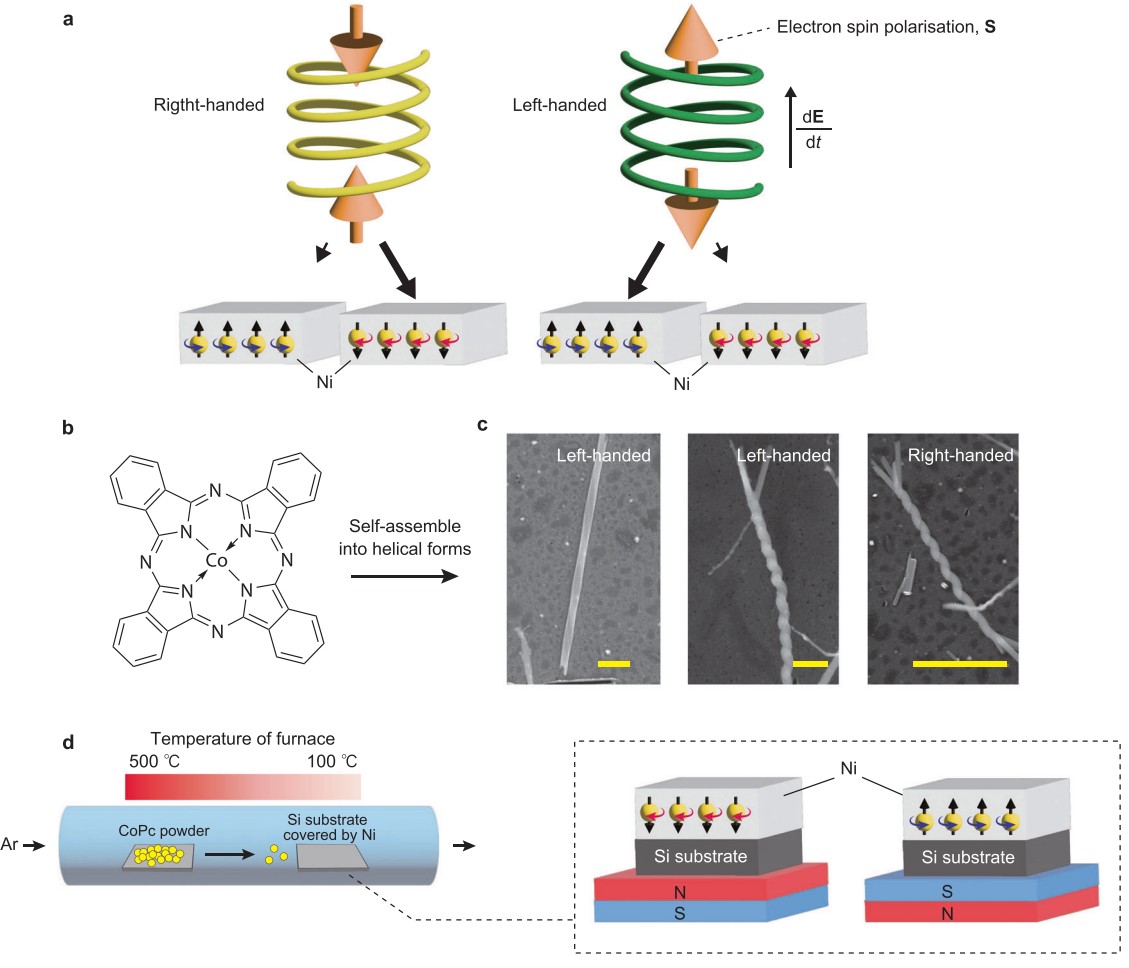

**Fig. 1 | Concept of CISS-based enantioseparation for CoPc helical supramolecules. a** A schematic illustration of emergent spins via CISS effect in chiral objects and its enantiospecific interaction with the magnetized substrates. The orientation of the anti-parallel spin pair is prescribed by the handedness of the system. For Ni spins, the horizontal (in blue or red) and vertical (in black) arrows describe the rotation and the spin direction, respectively. The black arrows pointing from the chiral molecules to magnetic substrates indicate the interaction between them. Longer arrows represent stronger interaction. **b** Molecular structure of the achiral Cobalt Phthalocyanine (CoPc). **c** Typical SEM images of synthesized helical suprastructures. Obtained helices have (Left) single-, (Center) double-, and (Right) even larger number of stranded structures. The yellow scale bars in each SEM image represent 1 μm. **d** Schematics of the physical vapor deposition (PVD) configurations for the test of CISS-based enantioseparation of helical supramolecules. The shaded red color represents the temperature profiles of the furnace (left and right sides are 500 and 100 °C, respectively). The powder of CoPc molecules heated at 500 °C is transferred to the right side by Ar flow, which is gradually cooled down to 100 °C to promote the crystallization of supramolecules. The synthesized supramolecules are captured by Si substrates covered by Ni. Note that our finite-element simulation on fluid dynamics has found no identification of any complicated motion of gas flow such as vortex motion in the PVD tube (see Supplementary Fig. 7 for more details).

symmetric $D_{4h}$ group, namely achiral (Fig. 1b)[22]. Furthermore, their planar structures are easily self-assembled into stacks via π−π interaction along the $b$-axis, giving rise to a strong tendency toward a one-dimensional (1D) superstructure[23]. In the case of cobalt phthalocyanine (CoPc), especially, even micro-meter-scaled helical-wire aggregations have been already achieved without introducing any chiral catalysis[24]. However, its enantioseparation has never been done as no practical method has been found. In the present study, stimulated by recent research on the enantioseparation by magnetic substrate, we have pursued the CISS-based enantioseparation for the helical aggregation of achiral CoPc, together with structural analysis by transmission electron microscopy (TEM) and electron tomography (ET). By synthesizing the helical supramolecules on the ferromagnetic substrates, we found that the preferred handedness of the helical supramolecules can be selected by the spin direction of ferromagnetic substrates, unambiguously verifying the relevance of CISS-based enantioseparation to the system. Remarkably, the helical supramolecule does not hold a continuous crystallographic twist but instead originates from sequential discontinuous changes in crystal axis, namely chiral crystal dislocations. The presence of spin-polarized current in such a helical supramolecule with discretized twists is substantiated by magneto-conductive atomic force microscopy (mC-AFM). These findings expand the application of the CISS effect from microscopic chiral structures to discontinuously twisting superstructures with mesoscopic length scale consisting of achiral components, having broader implications for a larger class of chiral-spintronic and optic devices.

## Results

### CISS-based enantioseparation by PVD method

To test the validity of the above-mentioned CISS-based enantioseparation, we first synthesized our target system, helical supramolecules of achiral CoPc, on ferromagnetic substrates through physical vapor deposition (PVD) technique, as schematically shown in Fig. 1d. We followed a synthesis scheme presented in ref. 24 although some of the conditions are slightly modified in order to adjust appropriately to our experimental setup (see also the "Methods" section and Supplementary Fig. 1 for details). Briefly, CoPc powder placed at the center of the upstream tube is heated at 500 °C. Then the evaporated molecules are transferred downstream by Ar flow, which is gradually cooled down to 100 °C to promote the crystallization of supramolecules. The synthesized supramolecules are captured by Si substrates whose surfaces are covered by ferromagnetic Ni and nonmagnetic Au with a thickness of 120 and 5 nm, respectively (Fig. 1d). The thin Au layer prevents Ni from being oxidized but does not diminish the magnetic or spin transport properties of Ni[13]. The easy axis of the Ni layer is in-plane, thus ensuring no out-of-plane magnetization in the absence of an external magnetic field[18]. With the condition, in-plane magnetic moments point to random directions, being canceled out on average, with which no CISS-based enantioseparation is expected[13,18,19]. As similar to the previous report[24], helical supramolecules with both right- and left-handed chirality have been obtained by our condition. Representative scanning electron microscopy (SEM) images of the supramolecules are displayed in Fig. 1c, showing a well-defined helical morphology. The synthesized helical supramolecules have typically a twist period of 100 nm–400 nm, with total length ranging from 1 to 10 μm. Such meso-scale helicity makes it possible to easily judge whether the supramolecule is the right- or left-handed by SEM observation. It is also noteworthy that our helical supramolecules have single-, double-, and even larger numbers of stranded structures (Fig. 1c), which indicates a contribution of screw dislocations to the twisting structure as discussed in literatures[25,26]. The microscopic structure of our helical supramolecules will be further investigated below.

We now explore the impact of ferromagnetic substrates on the crystallization of the left- and right-handed supramolecules. Samarium cobalt (SmCo) permanent magnets are placed underneath the substrates in order to set in up- or down-magnetized ferromagnetic moment of the substrates. The direction of the magnetic field is stuck along the out-of-plane direction due to the experimental constraint in our setup. The measured magnetic field generated by SmCo reaches ~ 300 mT, which is large enough to induce a considerable fraction of saturated magnetic moment of Ni (see Supplementary Fig. 2). The experiments were repeated either with SmCo magnets pointing with its North-pole up (N-pole) or with the South-pole up (S-pole), or without the magnets. We plot the ratio of right- and left-handed helices synthesized with these conditions in Fig. 2a; with S-pole up, the left-handed helical structure is preferentially obtained, whereas with N-pole up, is a right-handed one. Without SmCo magnets, the enantioselection seems absent, which is consistent with the fact that no out-of-plane magnetizations are expected in our Ni substrates. To quantitatively characterize the enantioseparation by ferromagnets, we here introduce an asymmetric factor, $g \equiv 2\{N(\text{Left}) - N(\text{right})\}/\{N(\text{Left}) + N(\text{right})\}$, where $N(\text{Left})$ and $N(\text{right})$ are the number of (or equal to the ratio of) observed left- and right-handed supramolecules, respectively. A roughly linear profile in the $g$ value against the magnetic field probes a coupling between the magnetization of the ferromagnet and selected chirality (Fig. 2c), in line with the perspective based on the CISS effect[13,18,19]. This scenario is further supported by our control experiments where only nonmagnetic Au is prepared on the substrates; there is no enantioselectivity both with and without SmCo magnets, reasonably ruling out a counter scenario that the external magnetic field or possible cyclotron motion of charged molecules induced by the magnetic field (not like the quantum exchange interaction from the ferromagnet) is a source of the observed enantioselectivity. Some asymmetries observed in the $g$ values between N- and S-poles, and also finite $g$ values even without the magnet (Fig. 2a and b) might be ascribed to some unexpected asymmetric interactions such as contamination of natural products made of homochiral amino acids and/or sugars, Coriolis force related to the earth's angular motion, or some asymmetric weak interactions (parity violation) existing in nature.

Our SEM images also found that most of the synthesized helical supramolecules lie down on the substrates, but it does not always mean that out-of-plane magnetization affects in-plane helical structures. One possible interpretation for this is that enantioseparation works at the onset of the adsorption of chiral nuclei onto the substrate, at which some of the nuclei would align perpendicular to the substrate. These nuclei are efficiently interacted with the magnetic substrate under the external out-of-plane magnetic field, resulting in the enantioseparation by the CISS effect. After this event, however, such nuclei that further grow up along out-of-plane direction become mechanically unstable, finally falling down on the substrate.

We also note that similar to the CISS effect, the so-called spinterface effect, a phenomenon that is tied to a broken inversion symmetry at the interface between molecules and metal, has the capability to produce spin polarization, possibly because of exchange and spin−orbit interactions[27,28]. However, the phenomenon is a polar effect and is not a chiral one, that is, the direction of spin polarization is not determined by the chirality of the molecules. Within this picture, the spinterface cannot be the leading factor to the enantioselection observed in our study, although it may help or enhance the CISS-based interactions.

Another important issue to be mentioned here is the role of a Co atom located at the center of the Pc molecule that has a localized spin of 1/2 stemming from $d$ orbitals and interacts with neighboring Co spin[22]. One may think that a possible formation of a chiral magnetic order of Co spins influences the enantioselectivity (ferromagnetic order has no chance to give enantioselection as we explained above based on the symmetrical consideration). However, the energy scale of the interaction is revealed to be at most 100 K[22,29], much smaller than

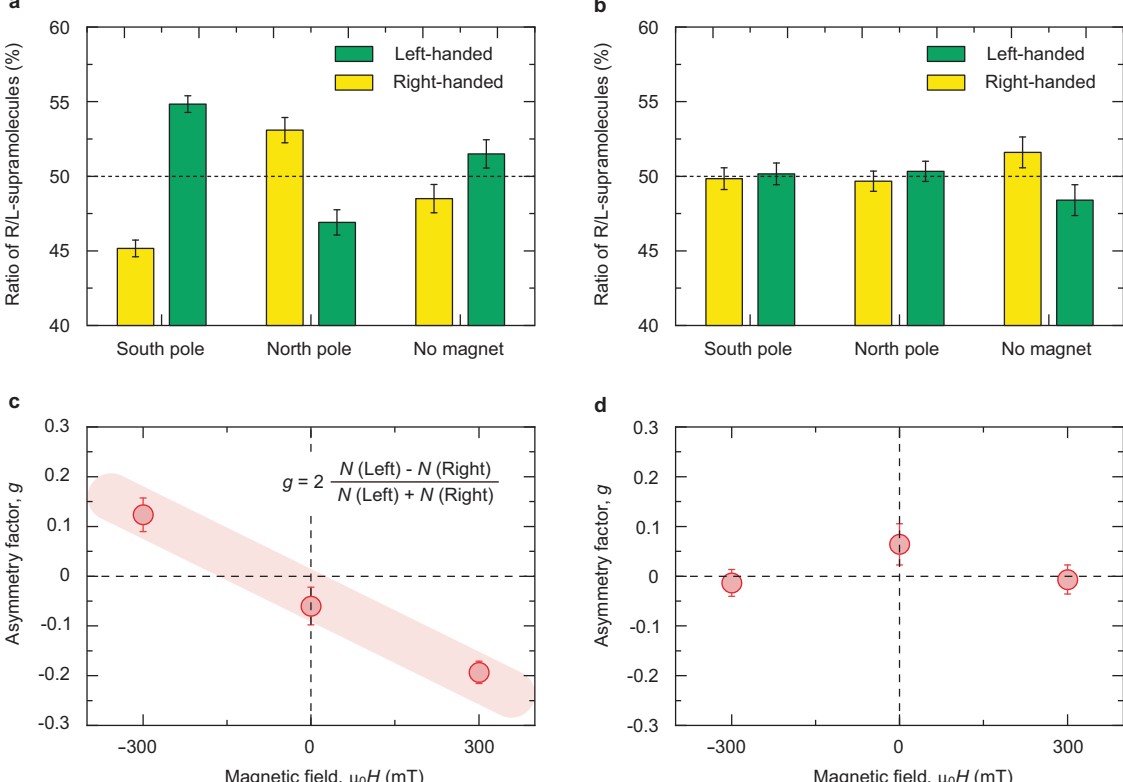

**Fig. 2 | Enantioseparation of supramolecular chirality by CISS effect. a, b** A ratio of the number of obtained left- and right-handed supramolecules with ferromagnetic substrates (**a**) and with nonmagnetic substrates (**b**), respectively. Upon constructing the plots in (**a**) and (**b**), the total number of approximately 600 helices was counted for each experimental condition to give statistically reliable results. The error bar in (**a**) and (**b**) is determined by the standard error. **c, d** Corresponding asymmetric factor, $g$, as a function of the magnetic field with ferromagnetic substrates (**c**) and with nonmagnetic substrates (**d**), respectively. The effect of the magnetic field emerges only for the synthesis condition with a ferromagnetic substrate. The error bar in (**c**) and (**d**) is evaluated based on the error bar used in (**a**) and (**b**).

the temperature we used for the synthesis. Thus, we can safely neglect the possible first-order contribution of the Co spin to our experiments, although the scenario that the amplitude of the anti-parallel spin pair is enhanced by the presence of Co spins still survives.

## Microscopic structure of helical supramolecules

Next, we address the microscopic structures of the helical supramolecules, focusing on a mechanism to create chirality from the achiral components. We investigate three-dimensional (3D) structural information by tomographic reconstruction from tilt series of two-dimensional TEM images. For this analysis, we selected a single-stranded helical structure to avoid a possible complexity in the cross-sectional image owing to overlapping supramolecules in double- or triple-stranded helix. Figure 3a shows a typical cross-sectional image of a single-stranded helical supramolecule, in which stripe patterns along the 1D-stacking axis are well-resolved, confirming the highly 1D-crystalline nature of the supramolecules. A clue for discussing the origin of helical nature is found in the fast Fourier transform (FFT) analysis of the cross-sectional image. In Fig. 3b and c, two pairs of discrete spots ($Q_1$ and $Q_2$) are clearly obtained in the FFT pattern of a yellow-rectangular region in Fig. 3a. The spots of $Q_1$ and $Q_2$ are not broadened along the azimuthal angle direction, indicative of an existence of two well-defined periodicities with different directions (see also line-profile analysis across $Q_1$ and $Q_2$ in Supplementary Fig. 3). The deduced periodicity in real space from the $Q_1$ and $Q_2$ spots is approximately 1.8–1.9 nm, roughly comparable to a diameter of a CoPc molecule. We also performed additional structural analysis based on electron diffraction images of the helix; the obtained lattice constants

differ from those of well-established superstructures such as α, β, and ε-phases[30,31], but seem to fit a structure of so-called *J*-phase[32] (see Supplementary Fig. 5 and Supplementary Table 1 for details). Here the simultaneous appearance of $Q_1$ and $Q_2$ can be interpreted as either a superimposed double-$Q$ structure or spatially separated multiple domains of a single-$Q$ structure. One useful way to distinguish these two possibilities is to examine $Q_1 + Q_2$ spots, which are allowed to appear only for the former double-$Q$ case. In Fig. 3b, although spots of higher order periodicity, $Q_3$ and $Q_4$, show up, $Q_1 + Q_2$ spots are not identified, excluding the former possibility. Furthermore, additional FFT analysis for the two restricted regions, positions 1 and 2, in Fig. 3a shows that either $Q_1$ or $Q_2$ spots appear in each position (Fig. 3d and e). These results lead us to conclude that our helical supramolecule includes spatially separated multi-domain structures characterized by different directions of periodicity. What is noteworthy here is that similar FFT results are obtained for different locations in the cross-sectional image (see Supplementary Fig. 4). Hence, the multi-domain nature is not a property that emerges only at a certain location but rather can be viewed as an overall characteristic of the helical superstructure.

Considering the well-separated $Q_1$ and $Q_2$ spots in the FFT pattern, the boundary of the multi-domains would be discontinuous, which is attributed to the formation of crystal dislocations[33]. As has been extensively discussed so far, in general, a screw dislocation combined with a point-like defect such as vacancies can work as a source of helix (which is referred to as helical dislocation), and such dislocations tend to interact with each other, resulting in the multi-stranded helix[25,26,34]. Indeed, we often encounter the multi-stranded helical structures as

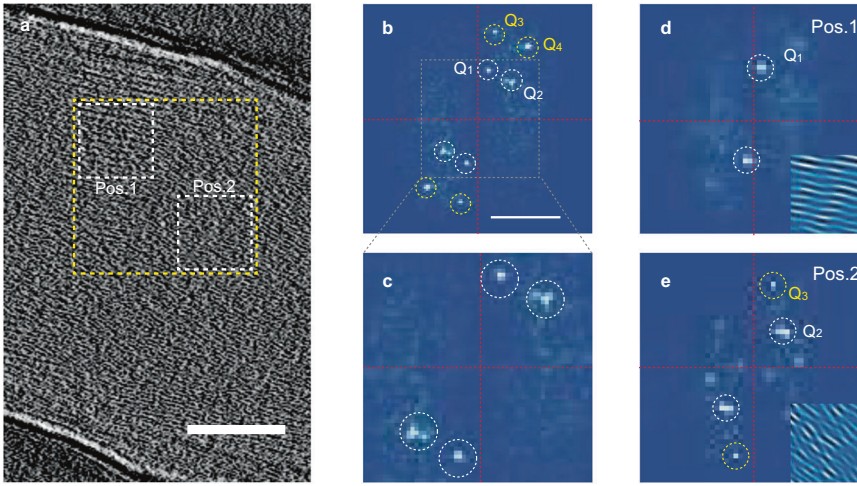

**Fig. 3 | Analysis of the TEM image of the helical supramolecule. a** A representative cross-sectional image of the reconstructed 3D topogram of the helical supramolecule. Clear stripe patterns are captured along 1D axis. The white scale bar represents 100 nm. **b** The fast Fourier transform (FFT) pattern of the yellow-highlighted area in (**a**), clearly showing several pairs of spots that are well separated from each other. White circles represent $Q_1$ and $Q_2$ spots, and yellow ones do higher order spots of $Q_3$ and $Q_4$. The scale bar represents $0.5\,\text{Å}^{-1}$. The resolution in the FFT pattern is estimated to be $6.5 \times 10^{-3}\,\text{Å}^{-1}$. **c** An enlarged image of **b** indicated by a gray-dashed line. **d**, **e** Additional FFT analysis for two squared regions in (**a**). FFT patterns of position 1 (**d**) and position 2 (**e**) exhibit either $Q_1$ or $Q_2$ spots, respectively. The insets in (**d**) and (**e**) show inverse FFT images restored from the corresponding pair of spots, displaying the periodic domains along noticeably different directions.

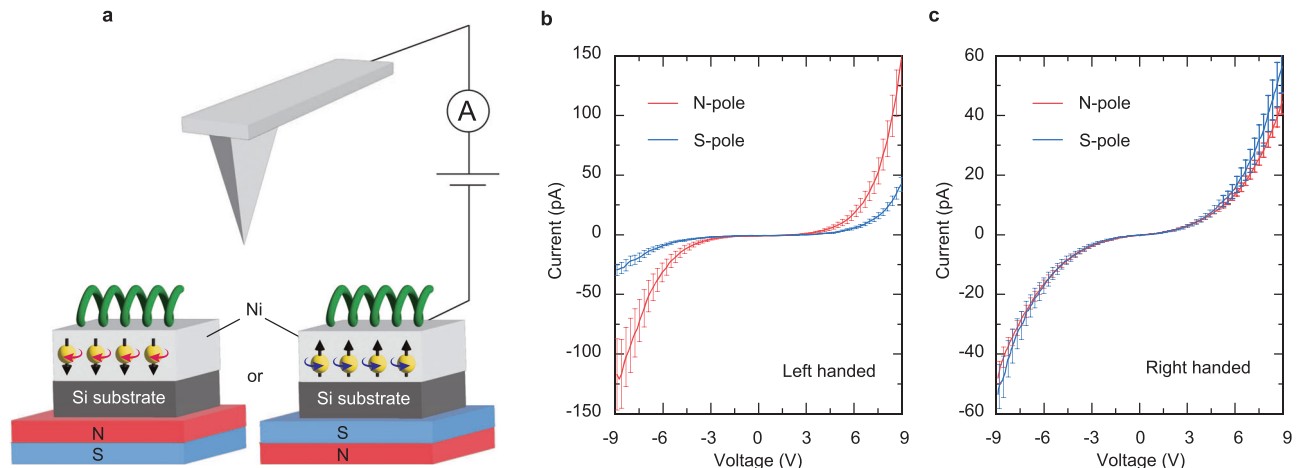

**Fig. 4 | Spin-polarized charge transport in helical supramolecules. a** Schematic illustration of mC-AFM measurements using a ferromagnetic substrate. **b**, **c** Current−voltage profiles for left- (**b**) and right-handed (**c**) helical supramolecules under the condition of N-pole up or S-pole up. At least 50 *I*−*V* curves were recorded and averaged for each handedness and magnetic field orientation. The standard error is adopted in (**b**) and (**c**).

shown in Fig. 1c, which shows good agreement with the notion of the helical dislocations. Based on these findings, it is very likely that the helical supramolecule stems from the assembled rod-like single domains demarcated by the helical dislocations. We note that there is no signature of a 1D line defect at the center of the helical supramolecule, meaning that the Eshelby twist, a continuous crystallographic twist associated with an axial screw dislocation in a 1D structure[35–38], is not pertinent to our helical structure.

### Generation of spin-polarized current in helical supramolecules
The remaining issue of importance is whether propagating electrons through such a discretized helical supramolecule can carry the CISS-induced spin as a spin-polarized current. One of the best approaches to check a spin-polarized charge transport is magneto-conductive atomic force microscope (mC-AFM)[39,40], where a ferromagnetic Ni layer serving as the bottom electrode is magnetized by a permanent magnet placed under the substrates (Fig. 4a). The orientation of magnetization is externally controlled by a motor connected to the magnet, making it possible to measure *I*−*V* profile of helical supramolecules at almost a same location with up and down magnetized conditions. In the experiments, the presence of spin-polarized current can manifest itself as a discrepancy in measured current between N- and S-pole up, as is similar to the case of tunnel magnetoresistance (TMR) effect[41,42]. We confirm that this is the case for our helical supramolecules as shown in Fig. 4b and c. The relationship between current with N- and S-pole up becomes opposite with different handedness of the supramolecules, highlighting that the emergent spin via the CISS effect points to an opposite direction depending on the handedness. The difference in current between two magnetized conditions in a left-handed supramolecule is much more prominent than that in a right-handed one, which may emerge from the differences in detailed properties of the supramolecule such as helical pitch, diameter, and total length. These

observations firmly validate the ability of spin propagation through the helical supramolecules with a mesoscopic length scale. Note that, in our setup, a screw axis of the helix is not parallel to the out-of-plane magnetized direction of the ferromagnetic substrate (Fig. 4a), but chiral symmetry also allows the spin-polarized current in the perpendicular direction against the screw axis[43].

## Discussion

The present results are summarized as follows: the helical supramolecules of achiral CoPc are enantiospecifically grown on the out-of-plane magnetized ferromagnet, and the magnetized direction switches the preferred handedness of the supramolecule. It straightforwardly means that the CISS effect works in the helical supramolecules, leading to the emergence of anti-parallel spin pair that is interacted with the ferromagnetic substrate. Importantly, our ET analysis revealed that the helical supramolecules do not possess continuous twisting morphology. Instead, a considerable fraction of the superstructure would be straight-like, and dislocations that are intermittingly located inside the supramolecule give the discretized twisting structure. Given that the constituent of the helix is achiral CoPc only, it is plausible that the chirality underlying the helical supramolecules is induced by the discretized twist at the dislocations. If this is the case, one expects that the CISS effect exhibits only at the dislocation points, and in other words, produces emergent spin dominantly at the points.

With this discussion in mind, such discretized twist structures with mesoscopic scale should be distinguished from conventional chiral systems such as chiral organic molecules or chiral inorganic crystals in that their chirality is determined by their microscopic compositional units. Thus, the successful application of CISS-based enantioseparation, which is based on the short-ranged exchange coupling with the substrate, to our helical supramolecules with mesoscale chirality emerging from the dislocation is no longer trivial and forces us to expand the limitation of CISS effect from the system with microscopic chirality to the one with mesoscopic chirality. It should be noted that this chirality is originating only from intermolecular interaction which is much smaller in energy than chemical bonding. One possible application based on the notion is a system of twisted van der Waals structures[38]. Recent development of bottom-up designing schemes of the layered van der Waals structures can tailor the well-defined twists with a wide range of length scale, but enantioseparation of them has not been achieved yet, although CISS-induced spin-polarized current has been reported[44]. We thus expect that a perspective based on our study would provide a facile basis to control the large-scale chirality of such systems and also design novel spintronic and optical devices.

It is widely accepted that crystallization is governed by two processes, nucleation, and growth[45], which also dominate the crystallization of our helical supramolecules. Then, an interesting open question is which part of crystallization the CISS-based enantioseparation affects. From our PVD experiments, obtaining insight into chiral selection at different periods of the crystallization process is difficult because of the lack of operando/in situ observations. Notwithstanding, we conjecture that a sizable chiral structure has been already grown when it reaches the substrates, and therefore a nucleation event of the chiral seeds, which happens far from the substrate, is not influenced by enantioselective field in our experimental setup. This may be a possible reason why the asymmetric factor obtained by our experiments is still small. Further experiments and theoretical studies are needed to study the impact of the chiral field on the enantioselectivity of chiral crystallization.

Finally, it is also instructive to compare the present observation with the previously reported absolute enantioselection by a combination of rotational and directional gravity[46]. In the study, the combination of purely physical fields generates a macroscopic chiral flow in a classical manner, successfully inducing absolute enantioselection of

helical supramolecules with porphyrin[46]. In contrast, our experiments here utilized a purely electromagnetic field in a quantum manner to prepare a chiral field, which is a complementary way with the enantioselection by physical field. These two examples strongly suggest the universality of the enantioselectivity based on the chiral influence that is independent of the detail of the fields.

## Methods

### Synthesis of helical supramolecules of CoPc by physical vapor deposition (PVD)

The helical supramolecules of CoPc were synthesized by using the physical vapor deposition (PVD) technique, similar to ref. 24. Our PVD setup includes a two-zone horizontal tube furnace with a diameter of 50 mm, and the temperature of each zone can be controlled independently, enabling us to finely tune the temperature both of the CoPc source and of Si substrates. The temperature profile of the furnace was measured with a thermocouple in advance. The detailed length scales of our PVD setup are summarized in Supplementary Fig. 1. First, the CoPc powder (2.0 g) and Si substrates, both of which are placed on quartz boats, were introduced at the upstream and downstream sides of the tube, respectively. Before the synthesis, the furnace was purged by a dry pump and Ar gas was to clean up the inside, and then Ar gas was kept to inject with a flow rate of 100 sccm during the deposition process. Next, furnace heaters are turned on to increase the temperatures of CoPc powder and Si substrates to 500 and 100 °C, respectively. The condition was maintained for 45 min for the deposition of CoPc supramolecules. In a cooling down process, we first turn off the furnace heater only for the substrate side, followed by an additional waiting time of 1 h. Finally, the other heater for the CoPc source is turned off to stop the synthesis, naturally cooling down the whole system to room temperature. The collected supramolecules on Si substrates are transferred to subsequent measurements.

### Preparation of substrates covered by ferromagnetic and non-magnetic metal

We used two different Si substrates in order to establish CISS-based enantioseparation. One includes thick ferromagnetic Ni capped by thin nonmagnetic Au with a thickness of 120 and 5 nm, respectively, and the other is covered by nonmagnetic Ti and Au with a thickness of 3 and 5 nm, respectively. They are prepared by sputtering or electron-beam evaporation technique.

To induce a magnetic field to the substrates during the PVD synthesis, we use samarium cobalt (SmCo) magnets, of which the magnetic field reaches about 300 mT, attached to the backside of the quartz boat by glue. Schematics of the substrates and SmCo configuration are displayed in Supplementary Fig. 1.

### Electron tomography

Helical supermolecules in ethanol were applied onto holey carbon film (Quantifoil R1.2/1.3 or R1/4, Quantifoil Micro Tools GmbH) over a 200 mesh copper grid (Maxtaform HF34), and the solution was dried up. The samples were first screened for electron diffraction with a JEOL JEM-2100 transmission electron microscope operated at an accelerating voltage of 200 kV. Rotational diffraction patterns were collected with an OneView camera[47] (Gatan, AMETEK) using SerialEM[48] and ParallEM[49,50]. For tomographic analysis, the samples were imaged at a specimen temperature of -93 K with a JEOL CRYO ARM 300 electron microscope[50] operated at an accelerating voltage of 300 kV. Tilt series of images were collected using SerialEM[48] with a modified version of a FastTomo script[51] in a bidirectional scheme from 0° to 60° and from −3° to −60° with a step of 3°. A nominal magnification was set to ×10,000, ×15,000, ×25,000 or ×50,000, and inelastically scattered electrons were removed through an in-column energy filter with an

energy slit width of 20 eV. At each tilt angle, dose-fractionated images were recorded on a K3 camera (Gatan, AMETEK) in CDS and counted mode at a dose rate of ~1 or ~2 e⁻/ Å².

Image stacks were drift-corrected, dose-weighted, and summed with the MotionCor2 algorithm[52] implemented in RELION-3.1[53]. Motion-corrected tilt series were aligned and reconstructed using AreTomo[54]. The handedness of the reconstruction was determined from tomographic reconstructions of left-handed Gold nanohelices (GATTAquant GmbH). Further characterization of the obtained 3D reconstructions was conducted using the software of UCSF ChimeraX and Gwyddion.

### Magneto-conductive atomic force microscopy (mC-AFM)

We performed spin-polarized conductive AFM measurements by using Dimension Icon XR (Bruker corporation). In this experiment, helical supramolecules were first dispersed in an ethanol solution, and then a small amount of the solution was dropped on the Ni (120 nm) and Au (5 nm) coated Si substrates. After the solvent is dried up completely, a thin Au layer with a thickness of 3 nm is evaporated to strengthen the attached supramolecules. If this process is omitted, the supramolecules are easily broken during the AFM scan. The Ni layer on the substrates works as the bottom electrode and is magnetized by a permanent magnet, whose maximum field is around 260 mT, placed under the substrates. A motor connected to the magnet is able to rotate the orientation of the magnet externally, making it possible to measure the $I–V$ profile of helical supramolecules at the same location with up and down magnetized conditions. Electric current is measured by a diamond-coated Si tip (CDT-FMR-SPL, NanoWorld), whose spring constant is 6.2 N/m, with sweeping the biased voltage in the range of ±9 V. The applied force between the tip and the samples is kept at 186 nN. At least 50 $I–V$ curves are measured for each condition, and the statistically averaged values are discussed in the main text. In Supplementary Fig. 6, SEM images of the measured helical supramolecules are shown.

### Magnetization measurements

The magnetization curves of our Ni film are characterized by using a superconducting quantum interference device (SQUID) magnetometer, MPMS-XL7 (Quantum Design). Large-sized Ni (120 nm)/Au (5 nm) film was deposited on the whole surface of the Si substrate (3 mm × 3 mm) for the measurement. A magnetic field was applied perpendicular to the plane of the film.

## Data availability

The raw data for mC-AFM are provided in the Source Data file. The SEM images generated in this study have been deposited in the Figshare database[55] (https://doi.org/10.6084/m9.figshare.23544648). Additional information related to this study is available from T.S. or H.M.Y. upon request. Source data are provided with this paper.

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

## Acknowledgements

We are grateful to the Equipment Development Center, Instrument Center (Institute for Molecular Science), and Advanced Research Infrastructure for Materials and Nanotechnology in Japan (ARIM Japan) under the Ministry of Education, Culture, Sports, Science and Technology (MEXT) for their technical support (Proposal Number JPMXP1222MS5015). This work was supported by JSPS KAKENHI (Grant No. 19H00891 to H.M.Y. and 20H01866 to T.S.), DAIKO FOUNDATION (to T.S.), JST-Mirai Program Grant Number JPMJMI20G5 (to K.Y.), and Research Support Project for Life Science and Drug Discovery (Basis for Supporting Innovative Drug Discovery and Life Science Research (BINDS)) from AMED under Grant Number JP22ama121006 (to K.Y. and S. M.-Y.). mC-AFM measurements were supported by Mr. Tadashi Ueda (Institute for Molecular Science).

## Author contributions

H.A. conceived the PVD synthesis. H.A., T.S., and T.M. performed mC-AFM measurements. S.M.-Y. and K.Y. performed ET experiments and K.Y., T.H., and K.T. calculated tomographic reconstructions. T.S. analyzed the dataset of mC-AFM and 3D topographic images. H.M.Y. supervised the project. T.S. wrote the manuscript in discussions with H.M.Y. All authors discussed the results and commented on the manuscript.

## Competing interests

The authors declare no competing interests.
