## [Peer Review File · Nature Communications]

REVIEWER COMMENTS

Reviewer #1 (Remarks to the Author):

This manuscript represents an important contribution to the field of enantioselective separation using as driving force the interaction between the molecule and a magnetic surface. The fact that the original structural components are achiral is a crucial ingredient to the novelty and significance of this work.

I only have three remarks that could be taken care of as a minor revision:

1. The manuscript contains the cryptic sentence "Without SmCo magnets, the enantioselection does not work." I think this requires further explanation, because it is not obvious how this affects the interaction with the magnetic surface.
2. The CISS effect is considered to be the dominant interaction responsible for the onset of chiral discrimination, however I think a more careful discussion of the possibility of exchange interactions to be partly responsible for the onset of enantioselectivity is important. Exchange and spin-orbit interaction are intrinsically coupled in a many-electron description of electron correlation and this should be discussed.
3. The authors do not discuss the possible importance of spinterface effects, that can be as important as the CISS effect.

Reviewer #2 (Remarks to the Author):

The manuscript presents interesting results showing enantioselectivity of helical super-molecule consisting of achiral cobalt phthalocyanines. This is a nice expansion of the work done presenting enantiomer separation and crystallization utilizing magnetic substrates and the CISS effect. Helical aggregation is conventionally directed either by its homochiral ingredients or by introduction of chiral catalysis. The work here exploiting enantiospecific interaction due to CISS, that has been known to work to enantio-separate a racemic mixture of chiral molecules, but for large assemblies. The observations mean that the application of CISS-based enantioselectivity can be expanded to mesoscopic chirality scale.

This may be important for applications supplying a simple large scale enantiomer separation technique.

I would happily recommend the publication of this article with minor corrections answering the following questions.

- 1) It is not clear to me how the out of plane magnetization induces the CISS effects on in plane assemblies. Please explain.
- 2) Is there an optimal angle for magnetization at which the separation is most effective?
- 3) What is the source for the asymmetry between north and south pole in Figure 2a and 2c?
- 4) Why do we see asymmetry in Figure 2b and 2d at zero magnetization?
- 5) Please explain the large difference between right handed and left handed spin polarization transport using mc-AFM figure 4.
- 6) I think that the authors should cite "Angewandte Chemie International Edition 59 1653 (2020)." and differentiate their work from these assemblies.

Reviewer #3 (Remarks to the Author):

The study by Yamamoto and co-workers reports the growth of 'supramolecular' helical fibres on a gold substrate by physical vapour deposition. Because the gold substrate is on top of a ferromagnetic nickel

film, which in turn is magnetised by an external magnetic film, there is an asymmetry in observed sign of helicity depending on the magnetization of the Ni film. In addition to the reported enantioselective assembly, asymmetric charge transport through the helical fibres is measured by contact with an atomic force microscope as counter electrode.

Because no new insight into CISS is provided I am not in favour of publication this work. The claim of novelty here is not substantiated: The authors state on page 3 " This novel enantioselectivity based on CISS effect has been already demonstrated to be able to separate a racemic mixture of helical amino acids, whose chirality is ab initio incorporated by their microscopic structures, into its two enantiomeric components". As achiral compounds often crystallize in chiral space groups or undergo spontaneous symmetry breaking in self-assembly, the claim that now this effect can also be seen for achiral compounds is mundane. Moreover, the report of asymmetric conductance as shown in Figure 4, is one of so many reported for all kinds of different systems.

Having said this, the presented manuscript does nothing add to existing knowledge.

Other points to consider:

It is actually not sufficiently shown that CISS is the working effect. The fibres are 3D materials with a thickness of good fraction of a micron. They are actually thin 3 D crystals (see Figure 3 TEM images, for example). As they contain therefore trillions of magnetic Co atoms, it has to be expected that exchange coupling makes these entities ferro magnetic. Consequently, the enantioselective 'crystal growth' is not necessarily related to the CISS effect. A small imbalance at nucleation plus amplification during growth would explain the asymmetry.

The non-magnetic experiment still shows an asymmetric result larger than the error. Hence there is a bias towards on chirality. What is the origin?

There is no asymmetry in electric conductance for the P-helices (Fig. 4c) with in the experimental error. What is the reason?

refs 16, 17, page range is missing

The use of language in the introduction is insufficient. Too many buzz words (spin-momentum locking, spin-fluctuations) or ill-defined statements. Certainly not directed to a general readership.

In conclusion, this is another study on top of the numerous papers about the CISS effect without providing new insight. I believe new studies should be designed such that new insight can be expected instead of adding more data.

Reply to the comments given by the Reviewer #1

[Comment 1-0]

This manuscript represents an important contribution to the field of enantioselective separation using as driving force the interaction between the molecule and a magnetic surface. The fact that the original structural components are achiral is a crucial ingredient to the novelty and significance of this work.

I only have three remarks that could be taken care of as a minor revision:

[Reply 1-0]

We are grateful to the Reviewer#1 for his/her thorough reading of our manuscript and illuminating suggestions, all of which helped us improve the manuscript so that our assertion gets more compelling. Furthermore, we are happy that the reviewer appreciates the importance and novelty of our observation of CISS-based enantioseparation of chiral nanofibers made of achiral components. In what follows, we reply to the reviewer's comments and explain the revisions we made. The revised parts are highlighted in red in the revised manuscript.

[Comment 1-1]

1. The manuscript contains the cryptic sentence "Without SmCo magnets, the enantioselection does not work." I think this requires further explanation, because it is not obvious how this affects the interaction with the magnetic surface.

[Reply 1-1]

Our Ni layer with a thickness of 120 nm has an easy magnetization axis parallel to the surface plane, which means that no out-of-plane magnetization is expected without an external magnetic field. However, CISS-based enantioseparation is only possible with out-of-plane-magnetized substrate as reported in many literatures (for example, Refs. 13, 18, and 19 in the revised main text). Therefore, we consider the enantioselection very unlikely without SmCo magnet that generates out-of-plane magnetic field, which is actually consistent with what we observed in Fig. 2 in the main text.

In fact, in-plane-magnetized substrate cannot be 'chiral' because it doesn't break mirror symmetry. One may think that the enantioselection can work for the molecules approaching to substrates in a shallow angle to the surface even without the magnetic field, in which case the molecule can feel Ni spins pointing to in-plane directions. In that case, however, the resultant enantioselection should be cancelled out by taking average of all magnetic domains, because

many possible in-plane orientation of the spins in the Ni layer would contribute to the result.

In the revised manuscript, we added the following explanation to clarify the above discussion in the 1st paragraph of “CISS-based enantioseparation by PVD method” section.

“The easy-axis of the Ni layer is in-plane, thus ensuring no out-of-plane magnetization in the absence of external magnetic field¹⁸. With the condition, in-plane magnetic moments point to random directions, being cancelled out on average, with which no CISS-based enantioseparation is expected^{13,18,19}.”

[Comment 1-2]

2. The CISS effect is considered to be the dominant interaction responsible for the onset of chiral discrimination, however I think a more careful discussion of the possibility of exchange interactions to be partly responsible for the onset of enantioselectivity is important. Exchange and spin-orbit interaction are intrinsically coupled in a many-electron description of electron correlation and this should be discussed.

[Reply 1-2]

We suppose that the reviewer’s comment is directed to the role of the exchange interaction among Co spins or between supramolecular nanowire and magnetic substrate. We agree on the comment that this is an important issue to be more carefully discussed in the manuscript. Regarding this issue, let us point out that the major interaction that create chirality in the nanowires is not a strong chemical bonding but is a weak intermolecular interaction that forms chiral dislocations. Therefore, it is not at all trivial that one can expect the same electronic spin responses from the chiral nanowire as one can see on chiral molecules [see for example, Ref. R1: A. Dianat, et al. *Nano Lett.* **20**, 7077–7086 (2020).]. Nevertheless, there is still a possibility that screw dislocation inside the nanowire can exhibit CISS effect that can be enhanced by exchange interaction with the substrate. Amplitude of anti-parallel spin pair can be also enhanced by exchange interaction in the presence of Co spins, which may play an additional role in efficiently giving the enantioselection. This may of course include many-electron effect and spin-orbit interactions.

Having the reviewer’s comment, we have noticed that we couldn’t appropriately emphasize the importance of the exchange interaction in our enantioselection based on the discussion above, and also it is instructive to mention the possible contribution of Co spin to reinforcing the enantioselection. We thus have largely revised the 4th and 5th paragraphs of “CISS-based enantioseparation by PVD method” section as described below, and Ref. R1 is newly added in a reference list as Ref. 28.

“We also note that, similar to the CISS effect, so-called spinterface effect, a phenomenon that is tied to a broken inversion symmetry at the interface between molecules and metal, has a capability to produce spin polarization, possibly because of exchange and spin-orbit interactions^{27,28}. However, the phenomenon is a polar effect and is not a chiral one, that is, the direction of spin polarization is not determined by chirality of the molecules. Within this picture, the spinterface cannot be the leading factor to the enantioselection observed in our study, although it may help or enhance the CISS-based interactions.

Another important issue to be mentioned here is the role of a Co atom located at the centre of Pc molecule that has a localized spin of 1/2 stemming from d orbitals and interacts with neighboring Co spin²². One may think that a possible formation of a chiral magnetic order of Co spins influences on the enantioselectivity (ferromagnetic order has no chance to give enantioselection as we explained above based on the symmetrical consideration). However, the energy scale of the interaction is revealed to be at most 100 K^{22,29}, much smaller than the temperature we used for the synthesis. Thus, we can safely neglect the possible first-order contribution of the Co spin to our experiments, although the scenario that amplitude of anti-parallel spin pair is enhanced by the presence of Co spins still survives.”

[Comment 1-3]

3. The authors do not discuss the possible importance of spinterface effects, that can be as important as the CISS effect.

[Reply 1-3]

To our best of knowledge, spinterface effect is in general a phenomenon that broken inversion symmetry at the interface between molecules and electrodes produces spin polarization. However, the phenomenon is a polar effect not a chiral one, that is, the direction of spin polarization is not determined by chirality of the molecules. Therefore, within this picture, the spinterface cannot be the leading factor to the enantioselection observed in our study, although it might be also possible enhance the CISS-based interactions.

Having the reviewer’s comment, we decided to add the discussion about the spinterface in the 5th paragraphs of “CISS-based enantioseparation by PVD method” section of the revised manuscript as described in the previous Reply [1-2].

Reply to the comments given by the Reviewer #2

[Comment 2-0]

The manuscript present interesting results showing enantioselectivity of helical super-molecule consisting of achiral cobalt phthalocyanines. This is a nice expansion of the work done presenting enantiomer separation and crystallization utilizing magnetic substrates and the CISS effect.

Helical aggregation is conventionally directed either by its homochiral ingredients or by introduction of chiral catalysis. The work here exploiting enantiospecific interaction due to CISS, that has been known to work to enantio-separate a racemic mixture of chiral molecules, but for large assemblies. The observations mean that the application of CISS-based enantioselectivity can be expanded to mesoscopic chirality scale.

This may be important for applications supplying a simple large scale enantiomer separation technique.

I would happily recommend the publication of this article with minor corrections answering the following questions.

[Reply 2-0]

We thank the Reviewer#2 for his/her thorough reading of our manuscript and illuminating suggestions, all of which helped us improve the manuscript so that our assertion gets more compelling. We also appreciate the reviewer for acknowledging our experimental study and for giving a high evaluation to this work. Below, we reply point by point to the comments and explain the revisions we have made. The revised parts are highlighted in red in the revised manuscript.

[Comment 2-1]

1) It is not clear to me how the out of plane magnetization induce the CISS effects on in plane assemblies. Please explain.

[Reply 2-1]

The reviewer's comment makes sense, so we here explain an expected process of enantioselection at the substrate in more detail. When nuclei of chiral supramolecules adsorb onto the surface of the substrate, their helical axes point to random directions but some of them would align perpendicular to the substrate. These nuclei are efficiently interacted with the magnetic substrate under the external out-of-plane magnetic field, resulting in the enantioseparation by

CISS effect. After this event, however, such nuclei that further grow up along out-of-plane direction becomes mechanically unstable, finally falling down on the substrate. This seems to be what happened in the adsorbing process in our PVD experiments.

To describe the process mentioned above, we added the following explanation in the 3rd paragraph of “CISS-based enantioseparation by PVD method” section of the revised manuscript.

“Our SEM images also found that most of the synthesized helical supramolecules lie down on the substrates, but it does not always mean that out-of-plane magnetization affects in-plane helical structures. One possible interpretation for this is that enantioseparation works at the onset of adsorption of chiral nuclei onto the substrate, at which some of the nuclei would align perpendicular to the substrate. These nuclei are efficiently interacted with the magnetic substrate under the external out-of-plane magnetic field, resulting in the enantioseparation by CISS effect. After this event, however, such nuclei that further grow up along out-of-plane direction becomes mechanically unstable, finally falling down on the substrate.”

[Comment 2-2]

2) Is there an optimal angle for magnetization at which the separation is most effective?

[Reply 2-2]

CISS-based enantioselection originates from the enantiospecific exchange interaction between emergent spin at the end of chiral supramolecule and the polarized magnetic spin in the Ni layer. Since the former spin is maximized along the helical axis of the molecule, the exchange interaction is the most enhanced if the helical axis is parallel to the magnetized direction of substrate. This situation gives an optimal angle for the enantioselection, which is in our case the angle perpendicular to the substrate because our external permanent magnet fixed below the substrate points only to out-of-plane direction. At present, the direction of external magnetic field cannot rotate in our setup, hindering an investigation of the angular dependence of enantioselection. Therefore, let us consider the study as a future work.

We newly mentioned above situation in the revised 2nd paragraph of “CISS-based enantioseparation by PVD method” section, as follows.

“Samarium cobalt (SmCo) permanent magnets are placed underneath the substrates in order to set in up- or down-magnetized ferromagnetic moment of the substrates. The direction of magnetic field is stuck along out-of-plane direction due to the experimental constraint in our

setup.”

[Comment 2-3]

3) What is the source for the asymmetry between north and south pole in Figure 2a and 2c

[Reply 2-3]

In general, the thermodynamic process for crystal nuclei formation is so complicated that some unexpected effects such as contamination of natural products made of homochiral amino acids/sugars, Coriolis force related to the earth's angular motion, or some asymmetric weak interactions existing in nature may play a minor role in inducing the asymmetry in g values. Based on the present study, unfortunately, we cannot present any clear answers about this issue.

In the revised manuscript, we added the following sentence in the revised 2nd paragraph of “CISS-based enantioseparation by PVD method” section to describe the possible origin of the asymmetry.

“... is a source of the observed enantioselectivity. Some asymmetries observed in the g values between N- and S-poles, and also finite g values even without the magnet (Figs. 2a and 2b) might be ascribed to some unexpected asymmetric interactions such as contamination of natural products made of homochiral amino acids and/or sugars, Coriolis force related to the earth's angular motion, or some asymmetric weak interactions (parity violation) existing in nature.”

[Comment 2-4]

4) Why do we see asymmetry in Figure 2b and 2d at zero magnetization?

[Reply 2-4]

This is also related to some unexpected effects explained in Reply 2-3. Although we carefully washed the glassware we used for the synthesis, some organic impurities may be introduced during our experiments, accidentally affecting the asymmetry in g values that sometimes surpass the error bars especially without the CISS-based chiral field. We are happy if the reviewer#2 also refers to the Reply 2-3.

[Comment 2-5]

5) Please explain the large difference between right handed and left handed spin polarization transport using mc-AFM figure 4.

[Reply 2-5]

As shown in the SEM images in Fig. 1c, our PVD setup produces large variety of helical supramolecules whose helical pitch, diameter, and total length vary from sample to sample. Since all of these properties quantitatively influence the amplitude of spin polarization, it is not unnatural to observe the large discrepancy between two samples used in the mC-AFM measurements. We thus focused only on qualitative difference in their *I-V* curves, namely the sign of the spin polarization. We mentioned this point in the 1st paragraph of “Generation of spin-polarised current in helical supramolecules” section of the revised manuscript.

“...depending on the handedness. The difference in current between two magnetized conditions in left-handed supramolecule is much more prominent than that in right-handed, which may emerge from the differences in detailed properties of the supramolecule such as helical pitch, diameter, and total length.”

[Comment 2-6]

6) I think that the authors should cite "Angewandte Chemie International Edition 59 1653 (2020)." and differentiate their work from these assemblies.

[Reply 2-6]

Thank you for the comment. Following the reviewer’s suggestion, we mentioned the reference in the introduction part and added it as Ref. 20 in the revised manuscript, as follows.

“This enantioselectivity based on CISS effect has been already shown to be able to separate a racemic mixture of helical amino acids, whose chirality is ab initio incorporated by their microscopic structures that can be coupled to the magnetic substrate via short-range exchange interaction, into its two enantiomeric components^{13,18–20} (Note that enantioselective electrochemical reactions are discussed in Ref. ²⁰).”

Reply to the comments given by the Reviewer #3

[Comment 3-0]

The study by Yamamoto and co-workers reports the growth of 'supramolecular' helical fibres on a gold substrate by physical vapour deposition. Because the gold substrate is on top of a ferromagnetic nickel film, which in turn is magnetised by an external magnetic film, there is an asymmetry in observed sign of helicity depending on the magnetization of the Ni film. In addition to the reported enantioselective assembly, asymmetric charge transport through the helical fibres is measured by contact with an atomic force microscope as counter electrode.

Because no new insight into CISS is provided I am not in favour of publication this work. The claim of novelty here is not substantiated: The authors state on page 3 " This novel enantioselectivity based on CISS effect has been already demonstrated to be able to separate a racemic mixture of helical amino acids, whose chirality is ab initio incorporated by their microscopic structures, into its two enantiomeric components". As achiral compounds often crystallize in chiral space groups or undergo spontaneous symmetry breaking in self-assembly, the claim that now this effect can also be seen for achiral compounds is mundane. Moreover, the report of asymmetric conductance as shown in Figure 4, is one of so many reported for all kinds of different systems.

Having said this, the presented manuscript does nothing add to existing knowledge.

[Reply 3-0]

First of all, we are grateful to Reviewer#3 for reviewing our manuscript in great detail and giving us illuminating comments. We respectfully receive the reviewer's comment that "*Because no new insight into CISS is provided I am not in favour of publication this work*" and subsequent comment that "*As achiral compounds often crystallize in chiral space groups or undergo spontaneous symmetry breaking in self-assembly, the claim that now this effect can also be seen for achiral compounds is mundane. Moreover, the report of asymmetric conductance as shown in Figure 4, is one of so many reported for all kinds of different systems.*". We feel that these evaluations are due to our insufficient introduction of our original paper. What we want to emphasize in this study is that *not the crystallization in chiral symmetry itself but the enantioseparation of chiral superstructure comprising only achiral ingredients* by CISS effect.

As reviewer#3 pointed out, the chiral crystallization from achiral components have been reported elsewhere so far. However, discriminating between the right- and left-handed chirality of such chiral crystal has been experimentally challenging. This is what our experiments demonstrated. Furthermore, the chirality produced by our PVD has a mesoscale helicity in origin,

which has never been considered as a platform of CISS effect in which chirality is based on the sub-nanometer scale structure of molecules. Therefore, our findings that CISS-based enantioselection works for such a mesoscale chirality is totally nontrivial.

Also, let us mention that in general chiral structures do not always exhibit the macroscopic properties reflecting their chirality. For example, CD spectrum, that is one of the most fundamental chiral responses, sometimes shows no signal even with the chiral compounds (For example, chiral molecular aggregates show CD active spectra only when the combination of the molecules is appropriate [Ref. R2]. Another example is intermolecular complex of Eu coordination complex which has both CD active and CD silent absorptions [Ref. R3]. These examples apparently mean that chiral intermolecular interaction does not always give electron's chiral response.). Considering these facts, it is also quite surprising that the macroscopic spin polarization appears through our supramolecules with the mesoscale chirality.

[R2] H. Zhu, et al, *Angew. Chem. Int. Ed.* **59**, 10868-10872 (2020).

[R3] M. Rancan, et al, *Cell Reports Physical Science* **3**, 100692 (2022).

In what follows, we reply point by point to the comments and explain the appropriate revisions of the manuscript. The revised parts are highlighted by red characters in the revised manuscript.

[Comment 3-1]

Other points to consider:

It is actually not sufficiently shown that CISS is the working effect. The fibres are 3D materials with a thickness of good fraction of a micron. They are actually thin 3D crystals (see Figure 3 TEM images, for example). As they contain therefore trillions of magnetic Co atoms, it has to be expected that exchange coupling makes these entities ferro magnetic. Consequently, the enantioselective 'crystal growth' is not necessarily related to the CISS effect. A small imbalance at nucleation plus amplification during growth would explain the asymmetry.

[Reply 3-1]

We can reasonably rule out the possibility that ferromagnetism inside the supramolecules is a source of enantioselection due to the symmetrical consideration as follows. In general, to discriminate between the right- and left-handed chirality by a physical quantity, the quantity must have the same symmetry as that of chirality. However, chirality and ferromagnetic spins should be classified into different symmetry because, with rotational operations, former is by definition unchanged but a sign of the latter changed. It can be also rephrased like this: a single spin or ferromagnet is not suitable to express molecular chirality because a sign of them changes with

rotational operations. Therefore, chirality cannot possess one-to-one correspondence with a single spin or ferromagnet. In contrast, anti-parallel spin pair as illustrated in Fig. 1a in the main text does not change under the rotational operations, thus being symmetrically allowed to correspond one to one with the chirality and to possess the enantioselective ability. These symmetrical discussion leads us to conclude that CISS effect and resultant anti-parallel spin pair are the only possible origin to induce enantioselection in our experiments.

In addition, the magnetic order of CoPc has been already examined both experimentally [M. Serri, *et al. Nat. Commun.* **5**, 3079 (2014)] and theoretically [W. Wu, *et al. Phys. Rev. B* **88**, 024426 (2013)], revealing that the exchange interaction is antiferromagnetic-like and the energy scale of the interaction is at most 100 K for α -phase and 2 K for β -phase. Obviously, no ferromagnetic order is expected at room temperature, making it very unlikely that the ferromagnetic order of Co spins is an origin of the enantioselection.

Finally, we also note that enantioselection may work if Dzyaloshinskii–Moriya interaction, if any, stabilizes a helical spin structure in the supramolecules. However, helical order can appear below the energy scale of the exchange interaction, and again is unrealistic to be realized in CoPc at room temperature giving that the exchange interaction is at most 100 K.

To describe the discussion above, we largely revised the introduction and results sections of the main text as follows.

2nd paragraph of the introduction part:

“It should be noted that the spin accumulation by CISS accompanies not a single spin but a pair of anti-parallel spins at the two ends of a chiral system¹⁶. The formation of latter is indispensable for the enantioselection because both chirality and the pair of anti-parallel spin pair can be classified into a same group of symmetry in that both of their signs are switched by space-inversion operation but not by rotational one, contrary to the case of a single spin or ferromagnetic spins whose signs become opposite by 180 ° rotation.”

5th paragraph of “CISS-based enantioseparation by PVD method” section:

“Another important issue to be mentioned here is the role of a Co atom located at the centre of Pc molecule that has a localized spin of 1/2 stemming from d orbitals and interacts with neighboring Co spin²². One may think that a possible formation of a chiral magnetic order of Co spins influences on the enantioselectivity (ferromagnetic order has no chance to give enantioselection as we explained above based on the symmetrical consideration). However, the energy scale of the interaction is revealed to be at most 100 K^{22,29}, much smaller than the temperature we used for the synthesis. Thus, we can safely neglect the possible first-order

contribution of the Co spin to our experiments, although the scenario that amplitude of anti-parallel spin pair is enhanced by the presence of Co spins still survives.”

[Comment 3-2]

The non-magnetic experiment still shows an asymmetric result larger than the error. Hence there is a bias towards on chirality. What is the origin?

[Reply 3-2]

This is the issue pointed out in Comments 2-3 and 2-4 above, so we would like the reviewer to refer to the Reply 2-3 and 2-4.

[Comment 3-3]

asymmetry

There is no asymmetry in electric conductance for the P-helices (Fig. 4c) with in the experimental error. What is the reason?

[Reply 3-3]

Although the observed asymmetry in I - V curves for P-helix is small, the difference in two I - V curves with N-pole and S-pole is still larger than the error bar especially at large voltage. Please look at the enlarged figure (Fig. R1). At present, the reason why the asymmetry in the left-handed helix is smaller than that in the right-handed is an open question, but the differences in helical pitch, diameter, and total length may quantitatively influence the amplitude of spin polarization. The comment raised here by reviewer#3 is closely related to the comment 2-5, so we are happy if the reviewer also read the Reply 2-5.

Hiving the reviewer's comment, we mentioned the situation above in the 1st paragraph of "Generation of spin-polarised current in helical supramolecules" section as follows.

“...depending on the handedness. The difference in current between two magnetized conditions in left-handed supramolecule is much more prominent than that in right-handed, which may emerge from the differences in detailed properties of the supramolecule such as helical pitch, diameter, and total length.”

Figure R1. A magnified image of the Fig. 4c at higher voltage range.

[Comment 3-4]

refs 16, 17, page range is missing

[Reply 3-4]

Thank you for your careful checking. We made corrections in accordance with the reviewer's suggestions.

[Comment 3-5]

The use of language in the introduction is insufficient. Too many buzz words (spin-momentum locking, spin-fluctuations) or ill-defined statements. Certainly not directed to a general readership. In conclusion, this is another study on top of the numerous papers about the CISS effect without providing new insight. I believe new studies should be designed such that new insight can be expected instead of adding more data.

[Reply 3-5]

Thank you for the comments regarding the buzz word. The suggested expressions such as spin-momentum locking, spin-fluctuations are common and familiar for us, but having the comments, we noticed that we should appropriately define them for a general readership. In the revised manuscript, we corrected all possible buzz words in the introduction part as follows.

“...chiral-induced spin selectivity (CISS) effect¹³⁻¹⁵; in chiral systems, electron’s charge motion engages with its spin angular momentum, the direction of which is symmetrically fixed to be parallel or anti-parallel to that of momentum depending on the handedness (namely, spin-momentum locking).”

“...which rearranges a charge distribution in the molecule, producing enantiospecific spin flow. In the closed system like molecule, the spin flow will be accumulated at the edge of the molecule.”

As for the latter comment, we respectfully say that there is a misunderstanding of the reviewer about the novelty of our work. Here, let us explain our emphasizing point again. The main achievement of this study is the first success of the enantioselection of mesoscale chiral superstructure consisting only of achiral components by utilizing CISS effect. So, chiral crystallization itself is not the goal of this paper at all. We believe that the revealed mesoscale chirality in the superstructure and emergent spin polarization in it are quite new and must be significant in that these findings expand the limitation of CISS effect from the system with molecular (or chemical bonding) chirality to the one with mesoscopic (or intermolecular) chirality. Since these points are beneficial for broader implications for a larger class of chiral-spintronic and optic devices, we believe in a high novelty of our work. Let us note that both of reviewer#1 and #2 are indeed appreciative of the novelty of our work. We revised the all parts of the manuscript to convey the achievements and novelty of our work more clearly, for example as follows.

2nd paragraph of the introduction part:

“However, a critical open question is whether this CISS-based enantioselectivity is applicable to chiral superstructure consisting only of achiral components where only weak intermolecular interaction is expected to work as the origin of its chirality. The issue is intimately tied to a possible absolute enantioselection via CISS effect.”

3rd paragraph of the introduction part:

“These findings expand the application of the CISS effect from microscopic chiral structures to discontinuously twisting superstructures with mesoscopic length scale consisting of achiral components, having broader implications for a larger class of chiral-spintronic and optic devices.”

1st paragraph of the discussion section:

“It straightforwardly means that CISS effect works in the helical supramolecules, leading to an

emergence of anti-parallel spin pair that is interacted with the ferromagnetic substrate.”

2nd paragraph of the discussion section:

“Thus, the successful application of CISS-based enantioseparation, which is based on the short-ranged exchange coupling with the substrate, to our helical supramolecules with mesoscale chirality emerging from the dislocation is no longer trivial, and forces us to expand the limitation of CISS effect from the system with microscopic chirality to the one with mesoscopic chirality. It should be noted that this chirality is originating only from intermolecular interaction which is much smaller in energy than chemical bonding.”

Finally, we again appreciate the reviewer #3's invaluable comments, which helped us notice that our original manuscript could not highlight our main selling point well. Now we improved the manuscript so that our assertion gets much clearer. We believe that our replies and revised manuscript overcome the concerns raised by the reviewer for publication in Nature Communications.

REVIEWERS' COMMENTS

Reviewer #1 (Remarks to the Author):

I am satisfied with the changes in the revised manuscript in response to my queries and comments. I think this is an important contribution to the field of induced enantio-selectivity involving the CISS effect.

I recommend now publication of this manuscript in Nature Communications.

Reviewer #2 (Remarks to the Author):

I feel that the authors have answered my questions and I am happy to recommend to publish the manuscript in Nature com.

Reply to the comments given by the Reviewer #1

[Comment 1]

I am satisfied with the changes in the revised manuscript in response to my queries and comments. I think this is an important contribution to the field of induced enantio-selectivity involving the CISS effect.

I recommend now publication of this manuscript in Nature Communications.

[Reply 1]

We are grateful to the Reviewer#1 for his/her thorough reading of our manuscript and giving a high evaluation to it.

Reply to the comments given by the Reviewer #2

[Comment 2]

I feel that the authors have answered my questions and I am happy to recommend to publish the manuscript in Nature com.

[Reply 2]

We are grateful to the Reviewer#2 for his/her thorough reading of our manuscript and giving a high evaluation to it.